# Accuracy at Lower Cost: Rethinking Client Selection in Federated Learning

## Abstract

Federated learning (FL) enables collaborative model training across multiple clients without sharing raw data, thereby ensuring privacy. A critical performance factor for FL is client selection. Under independent and identically distributed (IID) data, clients are chosen at random, which can lead to reduced accuracy, slower convergence, and higher communication cost. In this work, we present a systematic empirical study of client selection, revealing that random participation can significantly degrade performance. Motivated by these findings, we introduce a multi-objective optimization strategy that jointly balances model accuracy and communication cost under IID partitioning. For fast evaluation, we propose a dataset complexity-aware surrogate regressor that predicts the FL outcomes (e.g., accuracy or loss) for image classification tasks, thereby avoiding costly full model training. Using the predicted client configuration (number of selected and available clients) resulting from multi-objective optimization on a new dataset, and without requiring any additional training, our framework achieves 98.9% of the maximum attainable accuracy while incurring only 38.75% of the maximum communication cost. Moreover, it identifies a diminishing-returns regime that preserves 99.9% of peak accuracy while reducing cost to 63.12%. These results demonstrate that both the performance and variance of FL can be estimated solely by dataset complexity and client dataset size, enabling the identification of client configurations that best balance accuracy and communication costs.

## 1 Introduction

Artificial intelligence (AI) has rapidly advanced from explicit programming to data-driven models capable of learning from large datasets. However, challenges such as data scarcity, privacy concerns, and government restrictions on data sharing continue to limit the development and deployment of AI systems. Medical applications are particularly affected by these constraints. Clinical data, such as MRI and CT scans, contain sensitive patient information and are subject to strict privacy regulations, which restrict data accessibility for model training (Steeg et al., 2024; Elsheikh et al., 2024; Dumortier et al., 2022; Gitto et al., 2024; He et al., 2019). As a result, many studies identify insufficient data as a primary barrier to achieving high performance in medical machine learning (ML) applications. Approaches such as data augmentation (Fabian et al., 2021; Konidaris et al., 2019; Sun et al., 2020) and anonymization (Nohel et al., 2025; Fezai et al., 2023) have been explored to address this limitation and to enhance privacy preservation. These methods partially resolve the problem, as techniques for re-identifying anonymized data have also been developed (Steeg et al., 2024; Schwarz et al., 2019). Federated learning (FL), introduced by Google in 2017, has emerged as a promising solution by decentralizing training and keeping sensitive data local (McMahan et al., 2017; Konečný et al., 2016b;a; Xin et al., 2025). This paradigm allows for the construction of a global and high-performing model while preserving privacy. Nevertheless, FL introduces new challenges: training is resource-intensive, requires repeated communication rounds, and incurs high communication cost (Kairouz et al., 2021). Since communication cost scales with both the number of available and selected clients, it remains a major bottleneck in large-scale deployments (Zhu et al., 2024). The seminal Federated Averaging (FedAvg) algorithm (McMahan et al., 2017) shows that partial client participation can reduce communication without sacrificing accuracy, but it doesn't specify how to determine the optimal participation rate. Subsequent research has proposed adaptive and strategic client selection methods. It also reports there is a diminishing return beyond a certain

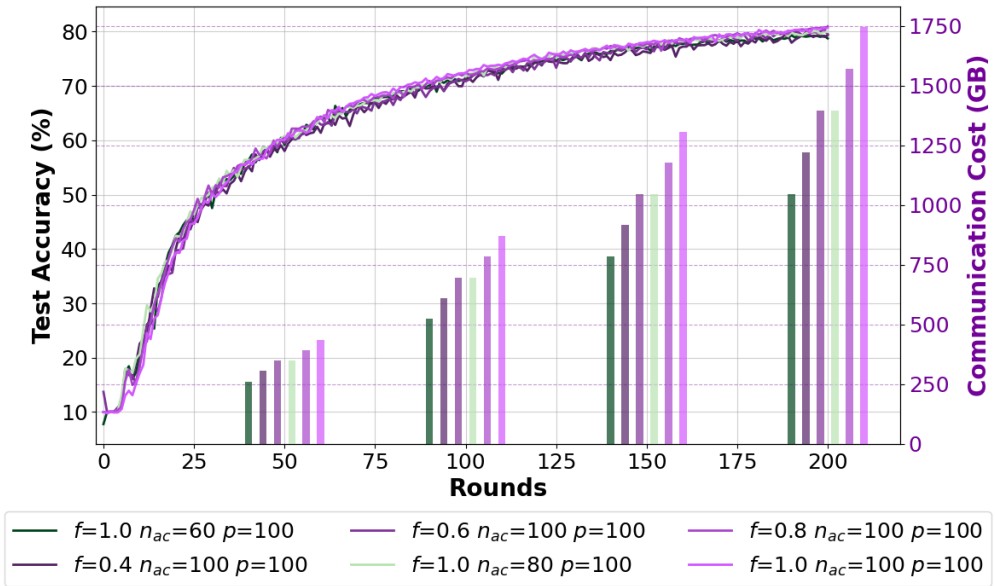

Figure 1: The impact of the number of available and selected clients on the trade-off between total communication cost and accuracy over rounds. Here, $n_{ac}$ denotes the number of available clients, and $f$ is the fraction of selected clients, so that the number of selected clients is given by $n_{sc} = f \cdot n_{ac}$. For better readability, the graph shows the test accuracy per round, while the bars represent the communication cost across the rounds.

point, such that adding more clients does not necessarily improve performance (McMahan et al., 2017). However, this threshold remains unknown. Furthermore, the optimal number of available and selected clients is largely unexplored, without finding an established method to determine the exact number required.

Figure 1 highlights a research gap summarized as follows: **under the same partitioning, different combinations of available and selected clients can achieve similar performance across rounds yet differ significantly in communication cost**. Although various combinations of available and selected clients may yield comparable accuracy over communication rounds, they often result in substantially different communication overheads. In some cases, doubling the number of participating clients produces nearly identical accuracy while almost doubling the communication cost. Hence, **increasing client participation does not necessarily enhance task performance and may instead introduce unnecessary communication overhead**. Note that higher communication costs have further implications beyond the volume of exchanged data. Increased communication cost leads to slower training, higher latency due to waiting for updates in each round, and greater energy consumption. Minimizing communication cost through random configurations, however, can degrade task performance, as shown in Figure 7.

This paper addresses this gap by **systematically analyzing the impact of client selection on FL task performance and communication efficiency**. Our paper advances the state of the art with the following contributions:

(i) to gain more insight about the impact of the hyperparameters on the FL task performance, an extensive experimental evaluation of hyperparameter effects is conducted, revealing diminishing returns in accuracy beyond a certain client threshold;

(ii) a bi-objective optimization formulation of client selection problem is proposed, where the decision variables are the number of selected and available clients, with objectives of maximizing task performance while minimizing cost;

(iii) to predict FL outcomes without training, dataset-complexity-aware surrogate regressor is built using ML models such as XGBoost, SVM, Decision Trees, and Adaboost;

(iv) to solve the proposed optimization problem, a bi-level optimization framework, that integrates a genetic algorithm with grid search, is introduced, identifying optimal client configurations.

## 2 RELATED WORK

Besides preserving privacy, one key objective of FL is to maximize model performance while minimizing the high communication cost incurred during training rounds. In each communication round, a subset of clients trains locally and sends model updates to the server, which aggregates them into a global model and broadcasts it back. communication cost increases with both the number of available clients and the number of selected clients to the aggregation, making it a major bottleneck in large-scale FL deployments (Zhu et al., 2024).

Early work by McMahan et al. (2017) introduced FedAvg, which demonstrated that partial client participation can maintain model quality while reducing communication costs, but did not propose a method to determine the optimal participation rate. Later, Yan et al. (2023) proposed *CriticalFL*, which identifies the critical learning period (CLP) during training. In this period, the number of participating clients is temporarily increased (e.g., doubled), before being reduced again in later rounds. This adaptive participation scheme improves accuracy compared to fixed client participation baselines while maintaining similar communication costs.

More recent work has shifted from random or full client participation to strategic client subset selection. Balakrishnan et al. (2022) introduced *DivFL*, which frames client selection as a submodular maximization problem to define clients whose gradient updates collectively represent the full client population. Specifically, DivFL defines a facility-location function over the clients' gradient vectors and employs a greedy selection algorithm to approximately maximize this function, offering provable approximation. Integrated with FedAvg, DivFL is theoretically analyzed for convergence under varying local steps, and in practice shows faster convergence, higher efficiency, and fairer performance across clients than random selection.

Fourati et al. (2023) introduced *FilFL*, which periodically applies a greedy filtering algorithm to select a subset of clients that maximizes a combinatorial objective. By assessing clients collectively rather than individually, FilFL improves efficiency, speeds up convergence, and boosts test accuracy by up to 10% in heterogeneous and time-varying settings.

Beyond optimizing which clients participate, another branch of research focuses on reducing the size or frequency of transmitted updates. For instance, Reisizadeh et al. (2020) proposed *FedPAQ*, which integrates periodic model averaging, letting clients carry out several local training steps before sending updates, with quantized message exchange to achieve substantial communication reduction, while still performing well on both convex and non-convex objectives.

Subsequently, the highlighted state-of-the-art works indicate that both optimizing client participation and designing efficient update transmission strategies are key to improving FL task performance, especially since the optimal number of participating clients is still unkown (Balakrishnan et al., 2022). Note that these studies focus **only** on optimizing the number of selected clients, always assuming all available clients are included. Therefore, this paper aims to advance the field of FL by strategically enhancing client participation by focusing on optimizing **both** the number of available clients and the number of selected clients to minimize the communication cost and maximize task performance.

## 3 METHODOLOGY: CONCEPT AND FORMULATION

In FL, involving more clients does not always lead to better task performance, as diminishing returns occur beyond a certain point. To address this challenge, we formulate our objective as an optimization problem in which the decision variables are $(n_{ac}, n_{sc})$, such that we can simultaneously **minimize communication cost** and **maximize task performance**. To unify these two objectives, the problem is reformulated as a multi-objective minimization task. Since this study is simulation-based (Beutel et al., 2020), the communication cost is defined as the total volume of data exchanged between the server and clients during training. This includes both the uplink communication, where selected clients transmit their local updates to the server, and the downlink communication, where the aggregated global model is broadcast to all available clients. The communication cost is defined

as follows:

$$\mathfrak{T}^{\text{cost}}(bits) \quad = \quad n_{sc} \cdot \#P \cdot R \cdot 32bits + n_{ac} \cdot \#P \cdot R \cdot 32bits$$
$$= \quad (n_{sc} + n_{ac}) \cdot \#P \cdot R \cdot 32bits$$

where the $\#P$ is the number of learnable parameters (weights and biases), $R$ is the number of communication rounds, $n_{ac}$ denotes the number of available clients, and $n_{sc}$ represents the number of selected clients.

The exchanged data between clients and server is based on the model learnable parameter transmitted during the communication round. The optimization problem is subject to the following constraints:

$$\text{s.t.} \quad n_{sc} \geq 2,$$
$$n_{ac} \leq P,$$
$$n_{sc} \leq n_{ac},$$

where $P$ denotes the number of data partitions, each corresponding to a potential client. The number of available clients $n_{ac}$ is constrained by the number of partitions, i.e., $n_{ac} \leq P$. The number of selected clients per round $n_{sc}$ must be at least two in order to ensure a FL setup ($n_{sc} \geq 2$), and cannot exceed the number of available clients ($n_{sc} \leq n_{ac}$). Moreover, if $f \in (0, 1]$ denotes the fraction of available clients selected in each round, the relationship between these quantities is given by $n_{sc} = f \cdot n_{ac}$.

### 3.1 DATASET-COMPLEXITY-AWARE REGRESSOR FOR TASK PERFORMANCE PREDICTION

As detailed in the previous section, the communication cost function is known and depends on the decision variables. However, the task training performance remains unknown. To address this, we require a function capable of estimating task performance without the need for training. Such a function should rely on the decision variables and remain generalizable to any new dataset. To this end, an extensive set of experiments is conducted in which a ResNet-18 model is trained on four different datasets (for more details see Table 1) under various configurations of $n_{ac}$, $n_{sc}$, and $P$, resulting in 380 design points. FedAvg is employed as the aggregation method with independent and identically distributed (IID) data partitioning. The resulting design points are used to train a task performance regressor with the dual objectives of (i) identifying an optimal regressor design and (ii) predicting task performance on previously unseen datasets. To achieve this, it is important to gain insight into the complexity of the dataset being learned. Therefore, this paper employs existing dataset complexity assessment methods, namely Cumulative Spectral Gradient (CSG) (Branchaud-Charron et al., 2019), Area Under the Laplacian Spectrum (AULS) (von Luxburg, 2007; Li et al., 2022), and Cumulative Maximum Scaled Area Under the Laplacian Spectrum (cmsAULS) (Li et al., 2022). These methods provide an approximation of how the classes in the dataset overlap, thereby revealing its complexity.

The task performance regressor uses dataset complexity, dataset characteristics, and decision variables ($n_{ac}$, $n_{sc}$) as input features (see Figure 11 illustrates the design of the regressor). In particular, the dataset characteristic refers to the number of training samples available to each client. Under an IID distribution, this relationship is given by

$$D_c = \frac{D_s}{P},$$

where $D_c$ denotes the number of training samples per client, $D_s$ is the total number of training samples in the dataset, and $P$ denotes the partitioning size. In total, 6 different regressor designs were evaluated: Linear Regressor, XGBoost, Decision Trees, Adaboost, Support Vector Machine (SVM), and Multi-Layer Perceptron (MLP). Task performance was assessed using two metrics, loss and accuracy, to determine whether either could be predicted more effectively. For validation, a new dataset, SVHN, was used; this dataset was not included in the training data of the regressor.

As this regressor is designed to predict the accuracy or loss of FL training, a fair evaluation is conducted by training 75 design points using ResNet-18 on the SVHN dataset with different configuration combinations of $n_{sc}$, $n_{ac}$, and $P$, each trained for 200 communication rounds. This data is used to evaluate and validate the robustness of the methods on a completely new dataset, as SVHN was not included during the training. Based on the $R^2$ score and the mean squared error (MSE), all dataset complexity assessment methods (AULS, CSG, cmsAULS) yield identical results. There is

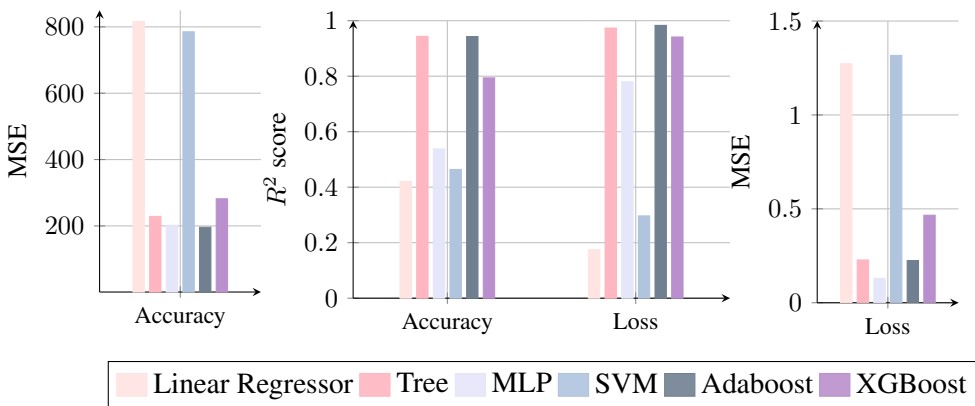

Figure 2: Validation results on a new dataset: Performance comparison (MSE and $R^2$) of various regressors for predicting loss and accuracy. The left graph compares the MSE of models when predicting accuracy. The middle graph shows the $R^2$ scores for predicting both accuracy and loss as performance. The right graph focuses on the MSE for predicting loss.

no observable task performance difference when using one complexity metric over another. Therefore, the focus will be shifted to determining which task performance metric is more predictable and which ML algorithm is better able to predict the task performance. Figure 2 shows three graphs presenting different evaluation perspectives of the regressors. The graphs are separated due to differences in scale, which improves readability and clarity. The overall conclusion across the three graphs shows, when predicting loss, the SVM and linear regression methods show poor predictive task performance. In contrast, when predicting accuracy, the MLP, SVM, and linear regression methods exhibit a marked reduction in performance on the new dataset, with an $R^2$ score below 0.6.

The results indicate that evaluating models using loss is more reliable than using accuracy. This is due to the loss function, based on cross-entropy, which considers the predicted probabilities rather than just the final label. Cross-entropy rewards the model predictions for being confident and correct, while heavily penalizing it when it is confidently wrong. In contrast, accuracy only measures the fraction of correctly classified samples, offering much less information.

Subsequently, the evaluation on the unseen SVHN dataset confirms that tree-based methods exhibit strong robustness and generalization capabilities, consistently maintaining high $R^2$ scores. In particular, Adaboost achieves the best performance, with $R^2 = 0.983$ and MSE $= 0.224$.

### 3.2 BI-OBJECTIVE OPTIMIZATION PROBLEM FORMULATION

After building, evaluating, and validating the regressors, we can predict the training loss based on our decision variables and the complexity of datasets that are not included in the training. The task performance prediction in our framework is based on the loss. When the optimization problem is defined in terms of loss, no reformulation is required. To ensure comparability, the loss is scaled to 100 so that it is not overshadowed by the larger magnitude of the communication cost. Similarly, the communication cost is normalized to the range [0, 100] for consistency. The resulting optimization problem, expressed using the loss formulation, is given by:

$$\min_{n_{ac},n_{sc}} \quad \lambda_1 \cdot (\mathfrak{T}^{\text{cost}}(n_{ac}, n_{sc}, R, \#P))^2 + \lambda_2 \cdot (100 \cdot \mathcal{L}(n_{ac}, n_{sc}, D_c, \mathcal{K}_{\text{data}}))^2$$
$$\text{s.t.:} \quad n_{sc} \geq 2,$$
$$n_{ac} \leq P,$$
$$n_{sc} \leq n_{ac},$$

Here, $\mathcal{L}$ denotes the predicted loss, and $\mathcal{K}_{\text{data}}$ denotes the dataset complexity, for which the CSG is used.

# 4 OPTIMIZATION PROBLEM SOLUTION AND RESULTS

The optimization problem balances communication cost and predictive loss using weighting factors $\lambda_1$ and $\lambda_2$. A bi-level strategy is employed, with an outer grid search over 169 $(\lambda_1, \lambda_2)$ pairs and an inner genetic algorithm that finds the optimal client configurations $(n_{ac}, n_{sc})$ under feasibility constraints. The communication cost is scaled relative to the worst-case traffic, while the predictive loss is estimated through a surrogate regressor model base on Adaboost. The final objective combines both terms in a quadratic form to ensure smoothness and comparability. Results provide the optimal $(n_{ac}, n_{sc})$ for each weighting pair, showing trade-offs between accuracy and communication cost depending on whether $\lambda_1 = \lambda_2$, $\lambda_1 > \lambda_2$, or $\lambda_1 < \lambda_2$. The contributions of this paper are illustrated in detail in Figure 3, which provides a comprehensive summary of the entire work.

Figure 4 illustrates the output of the bi-level optimization solution, where loss is used as the task performance metric and Adaboost serves as the regressor. By predicting the loss values for each $(n_{ac}, n_{sc})$ combination, it becomes possible to identify the point of diminishing returns, where increasing client participation yields minimal or negligible performance gains. To define the point of diminishing returns, we proceed as follows. The objective function, denoted by $\mathcal{F}$, captures the trade-off between communication cost and predictive task performance. The point of diminishing returns is identified as the configuration where communication cost starts to rise sharply while the relative gain in task performance drops below 0.1%. let $\mathcal{F}_{(1)} \leq \mathcal{F}_{(2)} \leq \cdots$ denote the objective functions sorted in ascending order of $\mathcal{F}^\star$. The first index $j$ that satisfies

$$\frac{\mathcal{F}_{(j-1)} - \mathcal{F}_{(j)}}{\mathcal{F}_{(j)}} < \tau,$$

with a tolerance of $\tau = 10^{-3}$ (i.e., 0.1%), marks the point of diminishing returns.

Each point in the figure corresponds to a unique $(n_{ac}, n_{sc})$ configuration. The proposed solution provides flexibility by allowing the selection of a target task performance level or a maximum allowable communication cost, from which the optimal values of $n_{ac}$ and $n_{sc}$ are derived. The optimal trade-off point occurs when $\lambda_1 = \lambda_2 = 1$, At this point, the loss and the communication cost are equally weighted, ensuring that neither objective dominates the other.

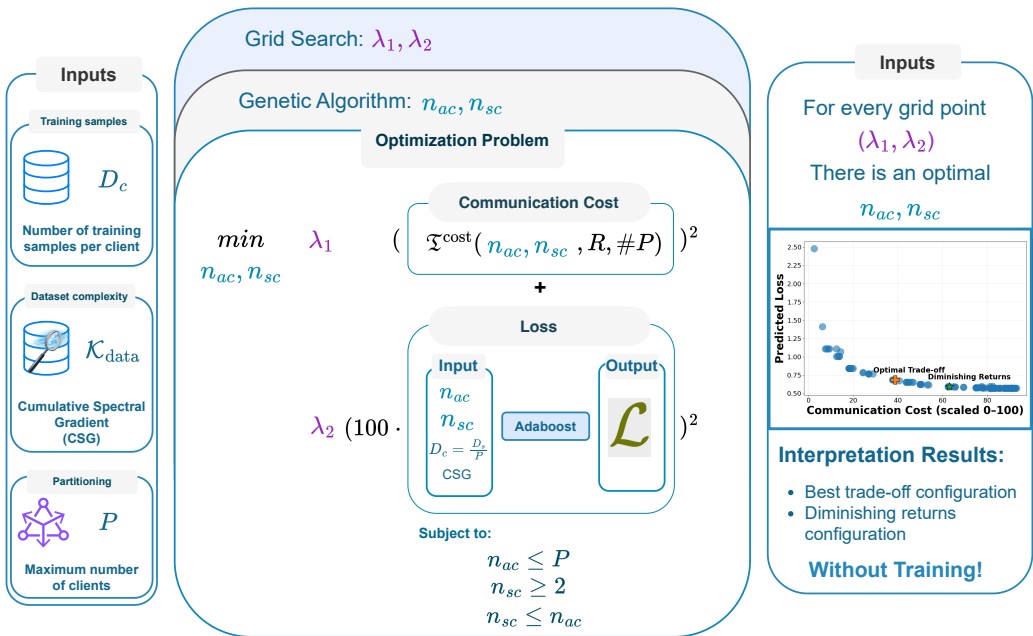

Figure 3: The proposed framework: our optimization-based method (no training required) predicts learning behavior from $D_c$, CSG, and $P$, and identifies client configurations $n_{sc}$ and $n_{ac}$ for the optimal trade-off and diminishing returns points

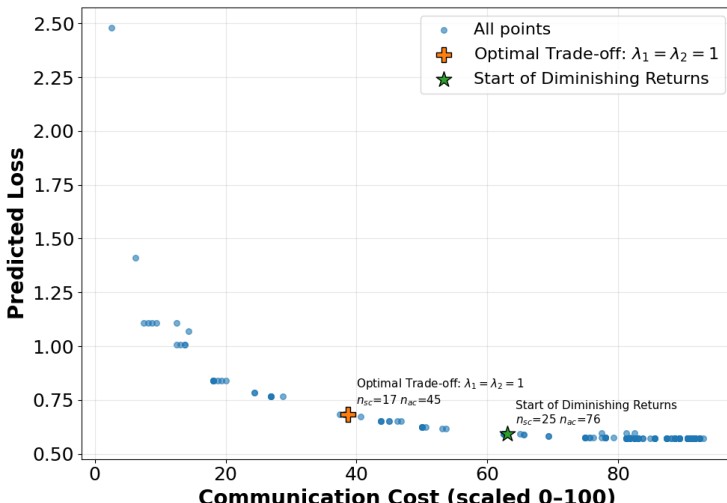

Figure 4: Results of the bi-level optimization solution. For each pair of weighting factors $\lambda_1$ and $\lambda_2$, an optimal tuple $(n_{sc}, n_{ac})$ is obtained. The corresponding predicted loss, estimated using Adaboost.

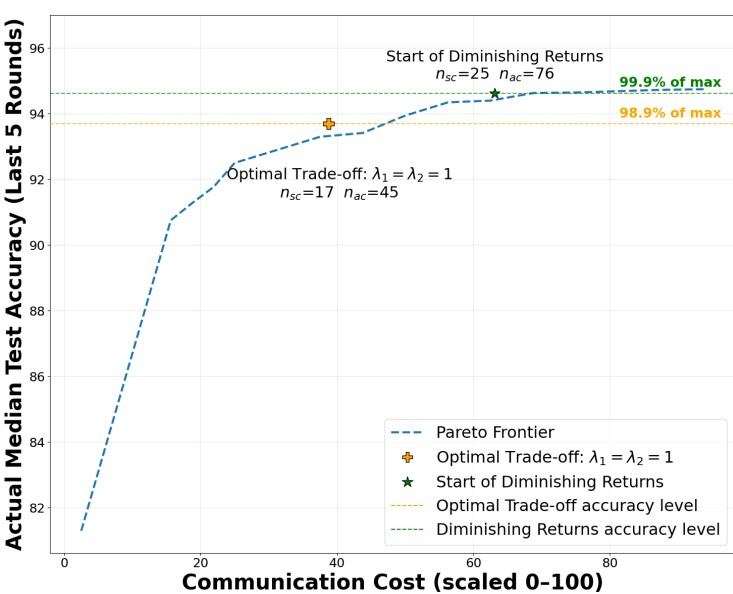

Figure 5: Validation of the bi-level optimization results. The resulting tuple $(n_{sc}, n_{ac})$ from the optimization problem is trained for 200 communication rounds using FedAvg as the aggregation method, with $P = 80$.

After obtaining the predicted $n_{ac}$ and $n_{sc}$, two configurations, the diminishing returns point and the optimal trade-off point, are validated to assess the robustness of the prediction. Validation is performed by training a ResNet-18 model in the FL setting for 200 communication rounds using the predicted configurations. The results, presented in Figure 5, plot the total communication cost (scaled to $[0, 100]$) on the $x$-axis against the actual median test accuracy on the $y$-axis.

The findings show that the optimal trade-off point achieves $98.9\%$ of the maximum accuracy while using only $38.75\%$ of the maximum communication cost. In contrast, the diminishing returns point achieves $99.9\%$ of the maximum accuracy while requiring $63.12\%$ of the maximum communication cost.

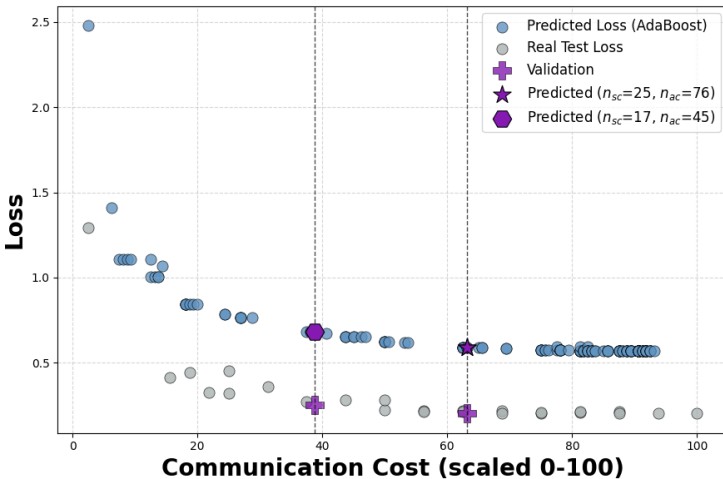

Figure 6: Predicted versus actual loss after training the resulting tuples $(n_{sc}, n_{ac})$ obtained from the bi-level optimization problem, evaluated on the SVHN validation dataset.

These results clearly demonstrate that the central research question of this paper is addressed: it is possible to determine the optimal client configuration $(n_{ac}, n_{sc})$ *without* performing any training in IID partitioning, identifying both the optimal trade-off and the point of diminishing returns.

The flexibility of the proposed framework allows the selection of $n_{ac}$ and $n_{sc}$ based on the specific requirements of the application. It is a general approach that does not require training the model, which significantly reduces training time while ensuring task performance. FL introduces more hyperparameters compared to centralized training, and the lack of publicly available datasets makes hyperparameter initialization and tuning even more challenging. This approach addresses these challenges by providing a predictive solution that helps guide such decisions. For clarity, the model predicts the loss based on a given set of inputs. Although this prediction achieves a high $R^2$ score, it is accompanied by a relatively high MSE. This occurs because the regressor can accurately capture the variation in loss but does not predict the exact actual loss that will be achieved. Instead, it predicts the upper bound of loss for the FL setting. Figure 6 shows the difference between the predicted loss and the actual loss, highlighting that the model estimates the loss upper bound rather than the exact value. Nevertheless, this level of prediction is sufficient to address the research question and to determine the optimal client configuration $(n_{ac}, n_{sc})$ without requiring any training.

## 5 DISCUSSION AND CONCLUSION

This paper proposes a new framework to determine the optimal client configuration that minimizes the communication cost and maximizes the task performance in IID data partitioning without requiring any training. The paper demonstrates that increasing the number of participating clients does not always translate into improved accuracy, as the effect of diminishing returns leads to higher communication costs without significant task performance gains. To address this challenge, we formulate FL as an optimization problem that aims to maximize model performance while minimizing communication overhead. To generalize our proposed approach, a dataset-complexity aware regressor was build to predict the loss based only on client configuration, training dataset size at clients and the complexity of the dataset. Experimental results indicate that near-optimal task performance can be achieved with substantially reduced communication costs: for instance, 98.9% of the maximum accuracy is attainable with only 38.75% of the communication cost, while 99.9% accuracy can be achieved at 63.12% of the maximum communication cost. Our findings can be perceived as the first attempt for exact determination of the client selection in FL.

## REPRODUCIBILITY STATEMENT

We have provided detailed descriptions of our methods, experimental setup, and hyperparameters in the main paper and the appendix to ensure reproducibility of the reported results. These details should be sufficient for researchers to reproduce our major findings. In addition, we will release the implementation code to further facilitate reproducibility.

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

## A   THE USE OF LARGE LANGUAGE MODELS (LLMS)

LLM and Grammarly were used to proofread the paper.

## B   APPENDIX

The following appendix provides supplementary material that supports and extends the main text of this paper. It relates to the design space exploration, showing that random configurations can degrade task performance and highlighting the existence of diminishing returns. With different partitioning sizes, it becomes evident that an optimal configuration exists for the trade-off proposed in this paper. The appendix also presents additional experimental observations for different dataset complexities, reinforcing the findings.

### B.1 Design Space Exploration for Federated Learning

This section investigates how varying key FL parameters, different combinations of $n_{sc}$ and $n_{ac}$, shapes the Pareto front when maximizing task performance while minimizing communication cost.

**Experimental protocol:**

A controlled procedure is followed: (i) vary one parameter at a time while keeping others fixed; (ii) train each configuration for 200 communication rounds; (iii) report task performance as the median test accuracy over the last five rounds; (iv) record the total communication cost (GB) for each configuration. The resulting design points are then used to construct and interpret Pareto-optimal trade-offs. For a better interpretation of the figures, Marker *shape* indicates the same $n_{ac}$ (available clients), and marker *color* indicates the same $n_{sc}$ (selected clients).

If $n_{sc} \geq n_{ac}$, a new random set of $n_{sc}$ clients is selected in each round. Figure 7 shows the design space exploration for CIFAR-10 with $P = 100$ using a convolution neural network and FedAvg as an aggregation strategy. Each point corresponds to a specific $(n_{ac}, n_{sc})$ configuration, plotted by its *median test accuracy* (y-axis) and *total communication cost* (x-axis). The Pareto front is overlaid to highlight configurations delivering the best task performance–communication cost trade-offs. The lowest-performing setup occurs when $n_{ac} = n_{sc} = 2$ with $P \gg 2$. This is the minimum configuration capable of running FL but yields the weakest task performance, the minimum viable configuration. Additionally, the observations indicate that increasing $n_{ac}$, $n_{sc}$ initially improves task performance; however, beyond a certain threshold, the benefits diminish, as further increases predominantly lead to higher communication overhead without proportional gains in accuracy.

### B.2 Pareto Front Variations under Different Partitioning size

This section investigates how varying data partitioning, where $P$ denotes the number of partitions such that a larger $P$ results in fewer training samples per client $(D_c)$, shapes the Pareto front when maximizing task performance while minimizing communication cost. The experiments highlight the clear impact of $P$ on learning task performance and its effect on communication cost (see Figure 8). Figure 8 highlights the differences between various partition sizes and compares them with centralized training. The legend indicates the fraction $f$, representing the proportion of selected clients relative to the total available clients, defined as $n_{sc} = f \cdot n_{ac}$ The results demonstrate that for larger values of $P$, the diminishing returns effect becomes evident. While increasing $n_{ac}$ and $n_{sc}$ initially improves convergence, the model performance saturates beyond a certain point. For each value of $P$, there exists an optimal tuple $(n_{ac}, n_{sc})$ that maximizes task performance while minimizing communication cost, reflecting an inherent trade-off. Additionally, the findings indicate that larger partition sizes generally lead to reduced task performance. Even when all partitions are utilized and all available clients are selected in each training round, task performance remains lower for larger partition sizes. Partition size directly determines the amount of data available to each client; when clients possess only limited data, increasing the number of clients does not necessarily yield improved accuracy. Collaborative learning is most effective when clients have sufficient data, a pattern clearly and explicitly demonstrated in this Figure 8. Increasing the number of partitions lowers the maximum achievable accuracy compared to both centralized training and FL with smaller partition sizes.

### B.3 Additional Experiment for Different Dataset Complexity

To provide deeper insight into the characteristics of the used datasets, Table 1 presents the most relevant dataset properties. Here, $dim$ denotes the image dimensions, $K$ represents the number of classes, $D_s$ indicates the total number of training samples, and the final column describes the content of each dataset. These characteristics are considered essential for summarizing a dataset. In addition, the table specifies whether the dataset's design points were included in the training of the regressor. As shown, the SVHN dataset is used exclusively for the validation of the *dataset-complexity-aware regressor*, meaning that the regressor has never been trained on this dataset. After defining the datasets used in this study, the focus shifts to estimating dataset complexity without performing training. Based on a review of the literature and to the best of current knowledge, three methods for assessing dataset complexity have been identified: cmsAULS (Li et al., 2022), AULS, and CSG (Branchaud-Charron et al., 2019). These methods represent the state-of-the-art

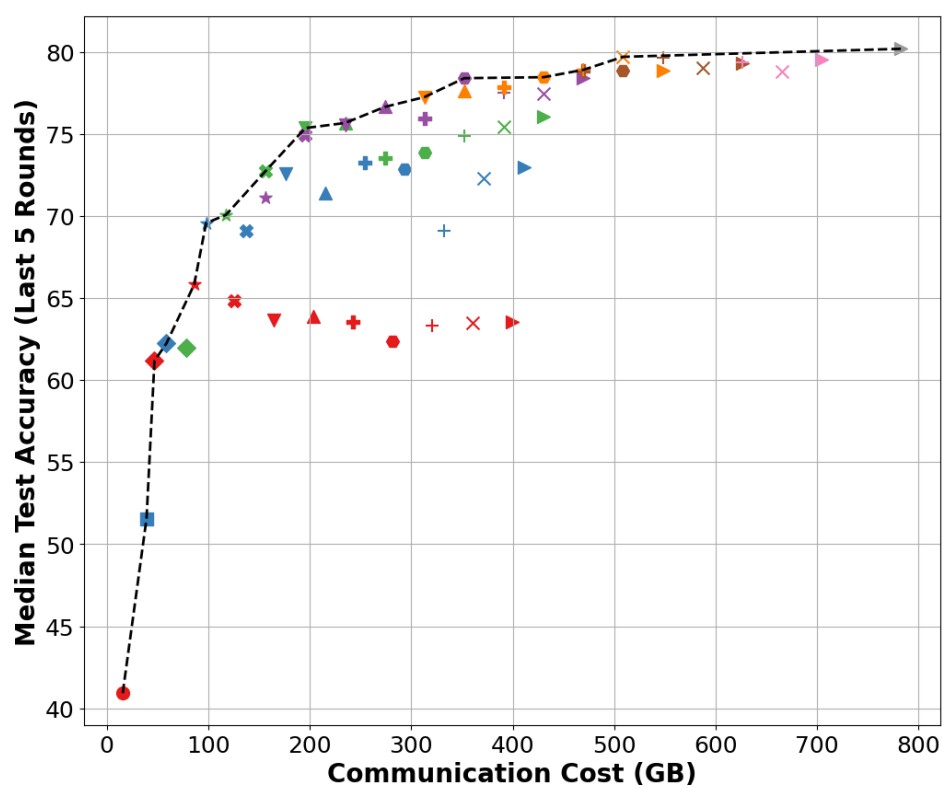

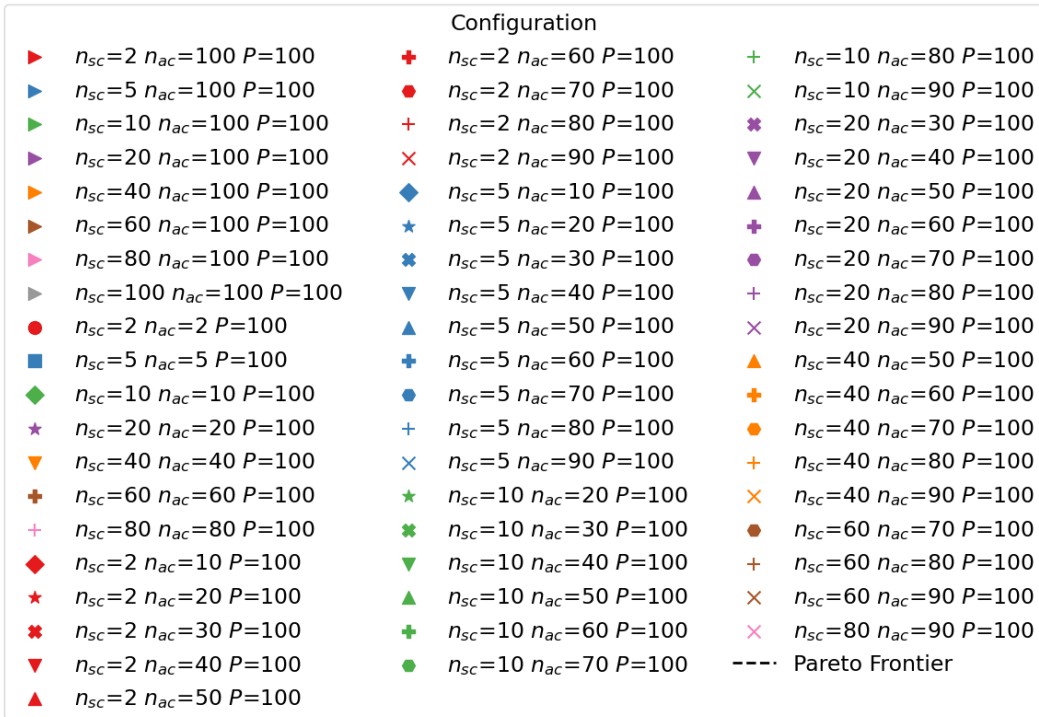

Figure 7: Design space exploration over $n_{ac}$ and $n_{sc}$ on CIFAR-10 ($P = 100$) using a convolutional neural network with FedAvg aggregation. The pareto front is indicated; each point reports the median test accuracy over the last five rounds after 200 communication rounds.

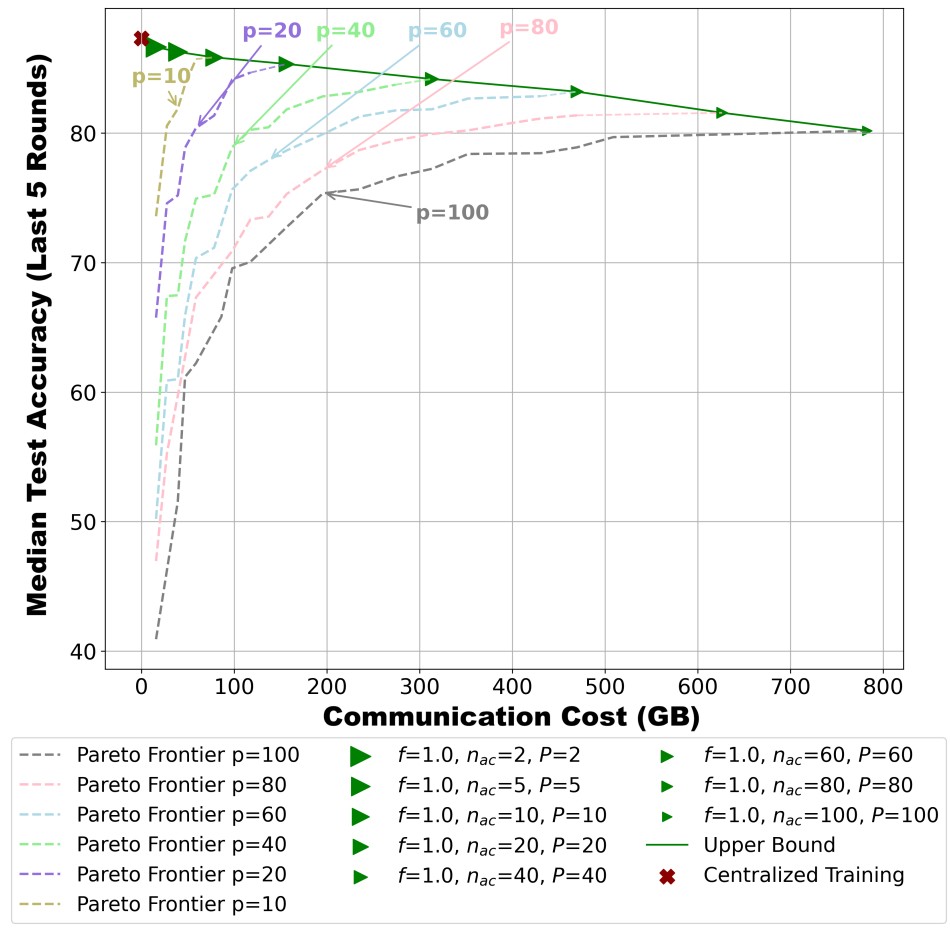

Figure 8: Pareto frontiers obtained under varying dataset partitioning size, illustrating the trade-off between task performance and communication cost.

Table 1: Characteristics of the used datasets (Branchaud-Charron et al., 2019).

| | | Dataset Characteristics | | | | |
|---|---|---|---|---|---|---|
| **Dataset** | **Training** | **Type** | **dim** | **K** | $D_s$ | **Content** |
| **MNIST** | ✓ | Grayscale | $28 \times 28$ | 10 | 50k | Hand written digits |
| **CIFAR-10** | ✓ | RGB | $32 \times 32$ | 10 | 50k | Various real images |
| **notMNIST** | ✓ | Grayscale | $28 \times 28$ | 10 | 18.7k | Printed digits |
| **SVHN** | | RGB | $32 \times 32$ | 10 | 73.3k | Printed digits |
| **STL-10** | ✓ | RGB | $96 \times 96$ | 10 | 5k | Various real images |

Table 2: The complexity of the datasets (Li et al., 2022).

| Dataset | Dataset Complexity Assessment Methods | | |
| --- | --- | --- | --- |
| | CSG | cmsAULS | AULS |
| MNIST | 0.045 | 0.144 | 0.675 |
| CIFAR-10 | 2.043 | 1.224 | 22.112 |
| notMNIST | 0.747 | 0.693 | 9.294 |
| SVHN | 1.826 | 1.100 | 20.142 |
| STL-10 | 3.546 | 1.914 | 49.134 |

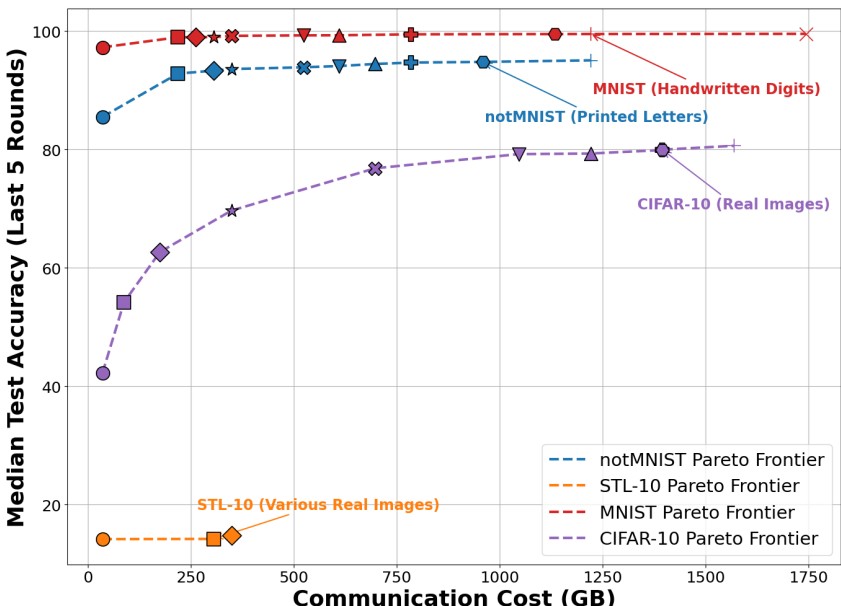

Figure 9: Pareto frontiers for $P = 100$ across the training datasets of the regressor.

in dataset complexity assessment, a comparative analysis is given in Li et al. (2022). However, there is no guarantee that these approaches will perform effectively in a FL setting. For this reason, all three methods are applied to construct separate regressors using different inputs, in order to determine which method is most suitable for the intended objective. The dataset complexity reflects the difficulty of distinguishing between two labels within the same dataset. In this study, STL-10 represents the most complex dataset, whereas MNIST, with a complexity score of $CSG = 0.045$, is the least complex. This also indicates that grayscale image datasets exhibit lower complexity than RGB datasets.

After presenting the different approaches for dataset complexity assessment, the next step is the construction of the regressor. A ResNet-18 model is trained in a federated manner on all training datasets under different configurations of $n_{ac}$, $n_{sc}$, and $P$. For each configuration, the training is conducted for 200 communication rounds, resulting in 380 distinct design points, which are then incorporated into the training process. For further interpretation, several observations were made after obtaining the design points. Figure 9 illustrates, for $P = 100$, the Pareto frontier for each dataset. These results are consistent with the properties summarized in Table 2 and Table 1, indicating that datasets with lower complexity tend to achieve higher task performance and that increasing the number of participating clients does not necessarily lead to task performance improvement. This trend is consistent with all previous experiments. Another important factor influencing task performance is the dataset size, as clearly demonstrated in the case of the STL-10 dataset. The impact of a small number of samples per client is clearly observed in the STL-10 dataset. For further clarification,

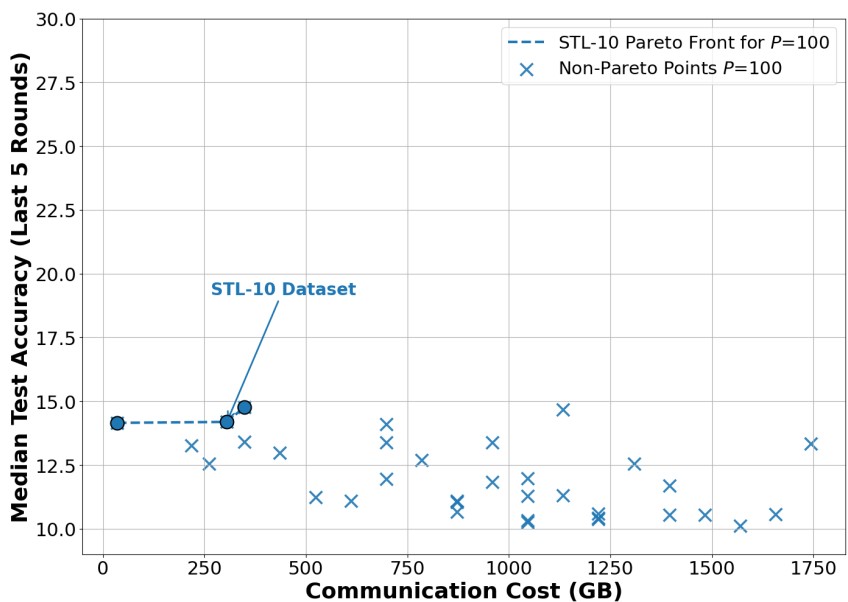

Figure 10: Design space exploration for STL-10 dataset with $P = 100$.

Figure 10 illustrates the identified issue: when clients have insufficient data, increasing the number of clients does not lead to task performance improvements in the system. This observation implies that, in FL, a minimum number of samples is required at each client for increasing the number of clients to be beneficial in achieving a given level of convergence. If this threshold is not met, the local models will fail to learn effectively, leading to poor task performance in the aggregated global model.

## B.4 THE REGRESSOR DESIGN

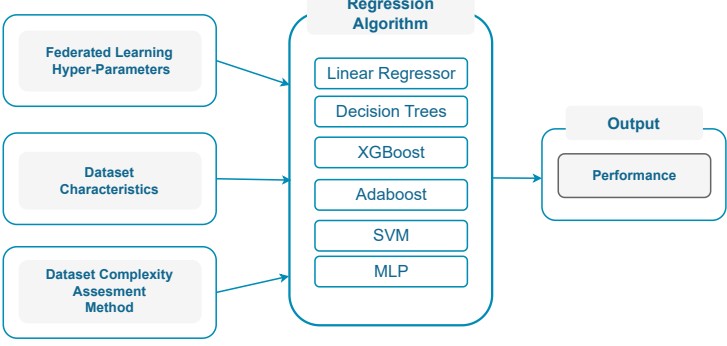

Figure 11: Illustration of the task performance regressor: inputs are federated learning hyperparameters, dataset characteristics ($D_c$ for trainable samples per client) and dataset complexity. Six regression algorithms are used to predict task performance, measured as loss or accuracy.

