# OpenReview forum: "Accuracy at Lower Cost: Rethinking Client Selection in Federated Learning"
_ICLR.cc/2026/Conference — ICLR 2026 Conference Withdrawn Submission_

### Official Review · Reviewer_D5f8 · 2025-10-19

**Soundness:** 3
**Presentation:** 2
**Contribution:** 2
**Rating:** 4
**Confidence:** 4

**Summary:**

The paper proposes a dataset-complexity-aware optimization framework for client selection in federated learning (FL), aiming to achieve high model accuracy at lower communication cost.

By leveraging surrogate regression models (notably AdaBoost) and multi-objective optimization via a bi-level (grid + genetic) approach, the method predicts FL performance without direct training.

Experiments across several image datasets (MNIST, CIFAR-10, STL-10, SVHN) demonstrate that the framework can achieve up to 98.9% of the maximum accuracy while using only 38.75% of total communication cost, suggesting strong cost–accuracy trade-offs.

**Strengths:**

- Introduces dataset-complexity-aware regressors to estimate FL outcomes without training, reducing computational overhead.
- Formulates client selection as a bi-objective problem balancing accuracy and cost.
- AdaBoost achieves high predictive accuracy ($R^2 = 0.983$, MSE = 0.224), showing that the surrogate model effectively approximates real FL performance.

**Weaknesses:**

- The approach is validated only on IID data, limiting its generalizability to real-world non-IID conditions that dominate practical FL settings.
- The cost function ignores latency, bandwidth variability, and client dropouts, which weakens its realism and applicability.
- Despite citing methods like DivFL, FilFL, and CriticalFL, the paper lacks quantitative benchmarking, leaving unclear how much improvement the method achieves over prior work.
- The surrogate regressor itself is pre-trained on numerous design points, so the framework still depends on substantial initial training effort.
- Figures (e.g., Figs. 3–5) lack axis clarity, and notation transitions are abrupt, which may hinder reproducibility and readability.

**Questions:**

- Please explain what the symbol “P” represents in Figure 1 to improve interpretability for readers unfamiliar with the variable.
- In Figure 1, the relationship between *selected clients* and the *fraction of selected clients (f)* should be kept consistent， for example, when $N_{ac} = 60$, corresponding $f$ values should be shown as 1.0, 0.8, and 0.6.
- The expression *“no training required”* could be misleading. It should be revised to “no retraining after surrogate pretraining” to more accurately reflect the methodology.
- Consider extending the surrogate regressor’s validation to larger and more heterogeneous datasets, such as FEMNIST or MedMNIST, to better demonstrate scalability and robustness.
- Including an analysis of system or platform latency would make the evaluation more realistic and strengthen the framework’s alignment with practical FL deployment scenarios.

---

### Official Review · Reviewer_uXdC · 2025-10-23

**Soundness:** 2
**Presentation:** 2
**Contribution:** 1
**Rating:** 2
**Confidence:** 5

**Summary:**

This paper investigates the trade-off between model accuracy and communication cost in Federated Learning (FL) under IID partitioning. The authors empirically observe that increasing the number of participating clients beyond a certain point yields diminishing returns. To address this, they propose a bi-objective optimization framework that jointly minimizes communication cost and predictive loss, where task performance is estimated via a dataset-complexity-aware surrogate regressor instead of full model training. The resulting framework predicts optimal client configurations (number of available and selected clients) that achieve near-maximum accuracy (e.g., 98.9% of the best accuracy) at reduced communication cost (38.75% of the maximum). Experiments across multiple datasets validate the predictive and optimization performance of the approach.

**Strengths:**

1. The study addresses a long-standing question in FL—how to determine the optimal client participation rate to balance communication efficiency and accuracy. This problem is practically important, especially for large-scale deployments.
2. Using a dataset-complexity-aware regressor to predict FL outcomes without actual training is an interesting idea, potentially saving substantial computation and communication resources.

**Weaknesses:**

1. The framework is built and validated entirely under IID settings. Since most practical FL scenarios involve non-IID data, the proposed model’s effectiveness and regressor generalization are uncertain. A non-IID evaluation is essential for broader relevance.
2. The paper reads more like an **engineering report or parameter study** than a scientific contribution. There is no rigorous reasoning, ablation, or insight into why particular relationships (e.g., diminishing returns) arise.
3. The communication cost is modeled linearly in terms of `nsc + nac`, ignoring factors like latency variance, asynchronous updates, and model compression. This oversimplification weakens realism.
4. While the paper claims the method “requires no training,” it trains multiple regressors and runs 380 full FL experiments to fit the surrogate model. This is computationally expensive. Thus, the framework does not actually save computation; it merely *shifts* the training effort offline.
5. There is no comparison with existing analytical or adaptive client selection frameworks, e.g., reward-based, fairness-aware, or reinforcement-learning-based methods. Without these, it’s hard to judge relative effectiveness.

**Questions:**

1. Can the surrogate regressor handle *non-IID* or *dynamic client participation*?
2. How many full FL runs are required to train the regressor, and what is the total computational cost?
3. How sensitive are the optimization results to the regression error and hyperparameter λ settings?
4. Could this method perform worse than simple random selection under non-IID conditions?
5. What happens if the surrogate mispredicts? does the optimization still produce reasonable client counts?

---

### Official Review · Reviewer_EmYg · 2025-10-23

**Soundness:** 2
**Presentation:** 3
**Contribution:** 2
**Rating:** 2
**Confidence:** 3

**Summary:**

This paper argues that, in federated learning (FL), final performance depends on the total number of available clients $n_{ac}$ and the number selected per round $n_{sc}$. It further analyzes a saturation effect: once client participation is sufficiently large, adding more clients no longer yields meaningful performance gains. Instead, additional participation increases communication costs. The paper therefore claims that determining an appropriate client count for FL participation is a critical issue that has been overlooked.
To support these claims, the paper presents preliminary experiments, evaluating various $(n_{sc}, n_{ac})$ configurations under the relation $n_{sc}=f \cdot n_{ac}$ where $f$ denotes the client selection ratio.
Finally, through various experimental designs, the paper demonstrates that the proposed method can identify a client count that achieves a target performance while explicitly accounting for communication overhead.

**Strengths:**

- The manuscript provides preliminary experiments to substantiate the underlying hypothesis and proposes a method to address the observed issues.
- It also supplies various experiments intended to support the proposed approach and to suggest that the resulting solution is near-optimal.

**Weaknesses:**

From this reviewer’s perspective, however, the paper has a major weakness. The proposed method is validated only empirically, without a theoretical analysis. As a result, it is difficult to justify the generalization of the conclusions to diverse FL settings.

In particular, the paper assumes all clients hold iid data. Under this assumption, increasing the number of clients effectively increases the amount of iid training data; once the data are sufficiently abundant, one expects diminishing returns resembling a logarithmic improvement curve. This observation aligns with well-known results on data scaling. However, in practice, the main challenge in FL is how to collaborate with clients under non-iid data distributions. Increasing the number of clients with non-iid data does not lead to a monotonic increase in global performance.

**Questions:**

This reviewer is curious whether there is a theoretical basis for predicting performance via regression without conducting actual training.

The regressor experiments should also vary both the training and test sets to enable broader validation. In the current experiment, the evaluation appears to rely on a training set (MNIST+CIFAR-10+notMNIST+STL-10) and a test set (SVHN), whose representativeness for generalization is questionable. The evaluation should test different combinations of training and test sets.

Moreover, the paper evaluates only ResNet-18. Given the diversity of modern architectures, and in the absence of theoretical guarantees, results derived from ResNet-18 do not necessarily transfer to other families such as Transformers.

Regarding the loss-prediction component, the paper employs AdaBoost and treats $n_{ac}$ and $n_{sc}$ as integers. It is not clear how the optimization problem in Section 3.2 is solved. Additional details on the optimization procedure (e.g., exact solver, relaxation, or heuristic) should be provided.

---

### Official Review · Reviewer_eh36 · 2025-10-31

**Soundness:** 2
**Presentation:** 1
**Contribution:** 1
**Rating:** 2
**Confidence:** 4

**Summary:**

This paper aims to solve a multi-objective optimization problem to find the optimal number of active and selected clients in FL. It starts off with the observation that increasing the number of clients does not necessarily improve the accuracy after a certain point, and only incurs more communication cost. Based on this, the authors provide a framework that allows to solve this multi-objective optimization problem to get the optimal number of active and selected clients in FL.

**Strengths:**

- The paper provides an interesting insight into FL where number of selected and active clients can actually be optimized for in the perspective of communication cost.

**Weaknesses:**

Overall, I had a hard time reading the paper due to its poor readability such as rather coarse figures, undefined variables (e.g., p in Figure 1,) and poorly defined variables (e.g., #P and P is a poor usage of defining variables). Moreover, I'm not really convinced that this can be a practical framework that can be deployed because i) calculating $D_c=D_s/P$ is highly improbable, ii) having iid distribution across clients, iii) being able to calculate the data complexity in a actual FL framework is highly impractical. I think this work can take some more time for improvement and polishing for publication or review.

**Questions:**

See Weaknesses.

---

### Author Response · Authors · 2025-11-26

**Dear Chairs and Reviewers,**


We would like to express our sincere gratitude for the time and effort you have devoted to review our submission.

Given the scope of the necessary changes, we respectfully request to withdraw the current submission. We intend to thoroughly revise the work and resubmit it in the future.

Thank you once again for your valuable feedback and understanding.


**Best regards,**
**The Authors**

---

### Note · Authors · 2025-11-26

I have read and agree with the venue's withdrawal policy on behalf of myself and my co-authors.